# Disproportionate impacts of COVID-19 on marginalized and minoritized early-career academic scientists

Hannah M. Douglas[1‡], Isis H. Settles[1,2‡], Erin A. Cech[3,4], Georgina M. Montgomery[5,6]*, Lexi R. Nadolsky[5], Arika K. Hawkins[7], Guizhen Ma[8], Tangier M. Davis[1], Kevin C. Elliott[5,7,9], Kendra Spence Cheruvelil[5,7]

1 Department of Psychology, University of Michigan, Ann Arbor, Michigan, United States of America, 2 Department of Afroamerican and African Studies, University of Michigan, Ann Arbor, Michigan, United States of America, 3 Department of Sociology, University of Michigan, Ann Arbor, Michigan, United States of America, 4 Department of Mechanical Engineering, University of Michigan, Ann Arbor, Michigan, United States of America, 5 Lyman Briggs College, Michigan State University, East Lansing, Michigan, United States of America, 6 Department of History, Michigan State University, East Lansing, Michigan, United States of America, 7 Department of Fisheries and Wildlife, Michigan State University, East Lansing, Michigan, United States of America, 8 Division of Social Sciences and History, Delta State University, Cleveland, Mississippi, United States of America, 9 Department of Philosophy, Michigan State University, East Lansing, Michigan, United States of America

‡ Co-first authors
* montg165@msu.edu

**Data Availability Statement:** The de-identified data associated with this manuscript is available on openICPSR. To protect participant confidentiality, the data file does not include demographic

## Abstract

Early research on the impact of COVID-19 on academic scientists suggests that disruptions to research, teaching, and daily work life are not experienced equally. However, this work has overwhelmingly focused on experiences of women and parents, with limited attention to the disproportionate impact on academic work by race, disability status, sexual identity, first-generation status, and academic career stage. Using a stratified random survey sample of early-career academics in four science disciplines ($N = 3,277$), we investigated socio-demographic and career stage differences in the effect of the COVID-19 pandemic along seven work outcomes: changes in four work areas (research progress, workload, concern about career advancement, support from mentors) and work disruptions due to three COVID-19 related life challenges (physical health, mental health, and caretaking). Our analyses examined patterns across career stages as well as separately for doctoral students and for post-docs/assistant professors. Overall, our results indicate that scientists from marginalized (i.e., devalued) and minoritized (i.e., underrepresented) groups across early career stages reported more negative work outcomes as a result of COVID-19. However, there were notable patterns of differences depending on the socio-demographic identities examined. Those with a physical or mental disability were negatively impacted on all seven work outcomes. Women, primary caregivers, underrepresented racial minorities, sexual minorities, and first-generation scholars reported more negative experiences across several outcomes such as increased disruptions due to physical health symptoms and additional caretaking compared to more privileged counterparts. Doctoral students reported more work disruptions from life challenges than other early-career scholars, especially those related to health problems,

information. [https://doi.org/10.3886/E172961V1] The full study dataset will be under an embargo up to 3 years following the end date of the grant funding (2023). Following this period, the non-identifiable data will be deposited with the Inter-university Consortium for Political and Social Research (ICPSR) at the University of Michigan. You may contact the corresponding author for more information and for data analysis code.

**Funding:** This research was funded by National Science Foundation ECR-EHR core research grants #2000579 (University of Michigan) and #1954767 (Michigan State University) awarded to Isis H. Settles, Kendra Spence Cheruvelil, Erin A. Cech, Georgina M. Montgomery, and Kevin C. Elliott. https://beta.nsf.gov/funding/opportunities/ehr-core-research-ecrcore The funders had no role in study design, data collection and analysis, decision to publish, or preparation of the manuscript.

**Competing interests:** The authors have declared that no competing interests exist.

while assistant professors reported more negative changes in areas such as decreased research progress and increased workload. These findings suggest that the COVID-19 pandemic has disproportionately harmed work outcomes for minoritized and marginalized early-career scholars. Institutional interventions are required to address these inequalities in an effort to retain diverse cohorts in academic science.

## Introduction

The COVID-19 pandemic has produced striking disruptions to the day-to-day work of academic scientists and has required many scientists to shift their time away from research and toward increased teaching, mentoring, and service [1–3]. However, accumulating evidence suggests that these negative pandemic-related outcomes are not shouldered equally. Early research suggests that the COVID-19 pandemic has exacerbated existing socio-demographic inequalities such as lower representation and lack of inclusion among faculty in academic science [4]. As a result, scholars have expressed concerns that the work disruptions of the pandemic will result in even greater long-term underrepresentation of people from marginalized groups in the academy [5–8].

Research to date on the inequitable impact of COVID-19 on scientists' work has overwhelmingly focused on women and parents in STEM, particularly in regard to research productivity as measured by the number of submitted research articles. Studies have found that compared to men, women faculty report increased workload from additional time spent engaged in teaching, mentoring, and service [4] and have experienced greater declines in research productivity (e.g., submitting fewer papers than men) [3, 4, 9–12], likely as a result of the greater service and caregiving demands they face. Among parents, it has been academic women, especially those with younger children, who have experienced the most significant declines in overall work time and their research productivity during COVID-19 [4, 11].

Yet, beyond these initial insights from studies of gender differences, we have an incomplete picture of the landscape of disadvantages caused by COVID-19. The potential disparate impact on other groups that experience social minoritization (i.e., underrepresentation) and marginalization (i.e., devaluation)–particularly along race and ethnicity, sexual identity, and disability status–have been largely overlooked. Given that faculty of color already have a greater workload from visible and invisible service demands [13], the combined toll of the COVID-19 pandemic and the social pandemic of racial injustice likely has exacerbated these inequities through additional student support and service work [14, 15]. The pandemic may similarly have disproportionately negative effects on scholars from other marginalized communities, such as sexual minorities, first-generation scholars, and scholars with disabilities. Research about work-related demands for people within multiple marginalized groups is critical because greater demands may be linked to the higher levels of stress, exhaustion, and burnout reported by women and faculty of color during the pandemic [3, 16–19].

Such divergent impacts are likely to be especially problematic for those early in their careers. One study found that psychological distress has increased among graduate students since the pandemic, with higher rates among students from marginalized groups [20]. Other research has found differences in the impact of COVID-19 by faculty rank. For example, assistant professors, who are more likely to have young children, have reported more substantial declines in research progress than tenured faculty [21]. Assistant professors have reported feeling insufficient support from their colleagues, a loss of networks and collaborations, and fears

about their ability to gain tenure [3, 15]. Because research on the impact of the pandemic on academic science has focused primarily on faculty ranks, we know little about the differential impacts of COVID-19 on academic work among different groups of early-career scholars (e.g., graduate students, postdoctoral scholars, assistant professors). COVID-19 related impacts on these early-career scholars are especially important for the academic pipeline, as experiences during these formative years may affect scholars' research trajectory and their interests in academic careers.

We addressed these knowledge gaps by examining differences in seven work outcomes due to the COVID-19 pandemic along six axes of socio-demographic variation (gender, parental caregiving, race, sexual identity, first-generation college student status, disability status) and three career stages. We conducted a representative survey of doctoral students, postdoctoral scholars, and assistant professors ($N$ = 3,243) using a stratified random sample of 124 departments of biology, physics, economics, and psychology in US institutions. Specifically, in April-May of 2021, we compared perceived changes over the past year in four work outcomes (research progress, workload, concern about career advancement, and support from mentors), as well as work disruptions due to three pandemic-related life challenges (physical health, mental health, and caregiving responsibilities) by each aforementioned socio-demographic status. Overall, we hypothesized that early-career scholars from marginalized groups would be more likely to report negative COVID-19 related work outcomes.

## Materials and method

Participants were doctoral students ($n$ = 2,687), postdoctoral scholars ($n$ = 335), and assistant professors ($n$ = 221) who completed an online survey administered in April and May 2021. Participants were recruited from four STEM fields (biology, economics, physics, and psychology) at 124 different departments that were randomly selected and stratified across 2011 National Research Council S-rankings of departments within each field (see *Participant Database* and S1 Table in S1 File for more details). Participants provided informed consent online by selecting 'Agree' on the informed consent page. If participants did not consent to the study procedures, they were redirected out of the survey. All study materials were approved and deemed exempt from full review by the institutional review boards at both the University of Michigan (HUM00193386) and Michigan State University (STUDY00004853).

We asked participants how the COVID-19 pandemic impacted their work during the past year across seven work outcomes. We asked about change in research progress, workload, concern about career advancement, and support from mentors (1 = *Greatly decreased* to 5 = *Greatly increased*). We also asked whether their work was disrupted in the past year by three COVID-19 related life challenges: physical health problems (e.g., sleep problems, headaches), mental health problems (e.g., mood problems, stress), and additional caretaking responsibilities at home (1 = *Did not disrupt my work at all* to 5 = *Disrupted my work a great deal*).

Participants self-reported their membership in seven socio-demographic statuses: gender (man, woman, or non-binary [including genderqueer or gender fluid]); race (underrepresented minority [URM; Black or African American, Hispanic or Latina/o/x, Middle Eastern or North African, Native American, American Indian, or other Indigenous group], Asian/Asian American, or White); sexual identity (sexual minority [lesbian, gay, bisexual, pansexual, queer, asexual, demisexual, or other] or heterosexual); being a first-generation college student; having a mental, physical, or learning disability; and career stage (doctoral student, postdoc, or assistant professor). For gender, because the number of individuals identifying as gender non-binary, genderqueer, and/or gender fluid was small and their responses to work outcomes tended to be in the same direction as women's responses, they were combined with women.

We also combined Black or African American, Hispanic or Latina/o/x, Middle Eastern or North African, Native American, American Indian, or other Indigenous group into URM due to their underrepresented status and stigmatization in the academy. See *SI Socio-demographic Variables* for additional details and S2 Table in S1 File for sample size of socio-demographic groups.

## Analytic plan

To assess differences in each work outcome described above, we ran seven separate multiple regressions on the full sample using the seven socio-demographic statuses as predictor variables: gender (women/non-binary or men), parental caregiving (primary caregiver, non-primary caregiver, or non-caregiver), race (URM, Asian, or White), disability status (at least one physical or mental disability or no disability), sexual identity (sexual minority or heterosexual), first-generation status (first-generation college student or not first-generation college student), and career stage (doctoral student, postdoctoral scholar, or assistant professor) (S3-S9 Tables in S1 File). Analyses controlled for academic field (biology, psychology, economics, or physics) and departmental ranking (tier 1, tier 2, or tier 3) (see *SI Participant Database* for details on covariates). All analyses were conducted using STATA 17 software.

Due to the larger size of the sample of doctoral students compared to postdoctoral scholars and assistant professors, we also ran each of the seven multiple regressions separately for doctoral students and a combined group of postdoctoral scholars and assistant professors (hereafter referred to as "postdoc/asst prof"); we included these two groups in the same subgroup analyses due to their smaller sample sizes and because participants in both groups are more advanced among early-career scholars. The subsample analyses were identical to those for the full sample except that for the doctoral student analyses, academic rank was not included and for the postdoc/asst prof analyses, academic rank compared postdoctoral scholars and assistant professors. The results of each set of regressions (full sample including all three career stages, doctoral student subsample, and postdoc/asst prof subsample) are described below; we first describe findings that are consistent across the full sample and both subsamples, and then highlight when there were subsample differences in the patterns of relationships.

To confirm that the variability in COVID-19 outcomes were not driven by the distribution of participants in departments, we ran seven additional multilevel mixed-effects regression models with respondents nested in departments. Department-level variation did not significantly account for the variability in any of the seven work outcomes; results for these models are in the (S10 and S11 Tables in S1 File). Finally, we conducted supplemental multiple regressions to capture the negative impact that COVID-19 disruptions had on job satisfaction, professional role confidence, turnover intentions, burnout disengagement, and burnout exhaustion; these are presented in the supplement for the overall sample as well as for the doctoral student and postdoc/asst prof subsamples (see S12—S16 Tables in S1 File for results).

## Results

Means, standard deviations, and correlations among COVID-19 impact variables are presented in S17 Table in S1 File for the full sample and each subsample.

### Work disruptions due to COVID-19

**Change in research progress (Fig 1).** Controlling for all other socio-demographic variables, we found that all participants, regardless of group membership, reported decreased research progress. Across samples, individuals with a disability reported a greater decrease in their research progress due to the pandemic compared to those without a disability. In the full

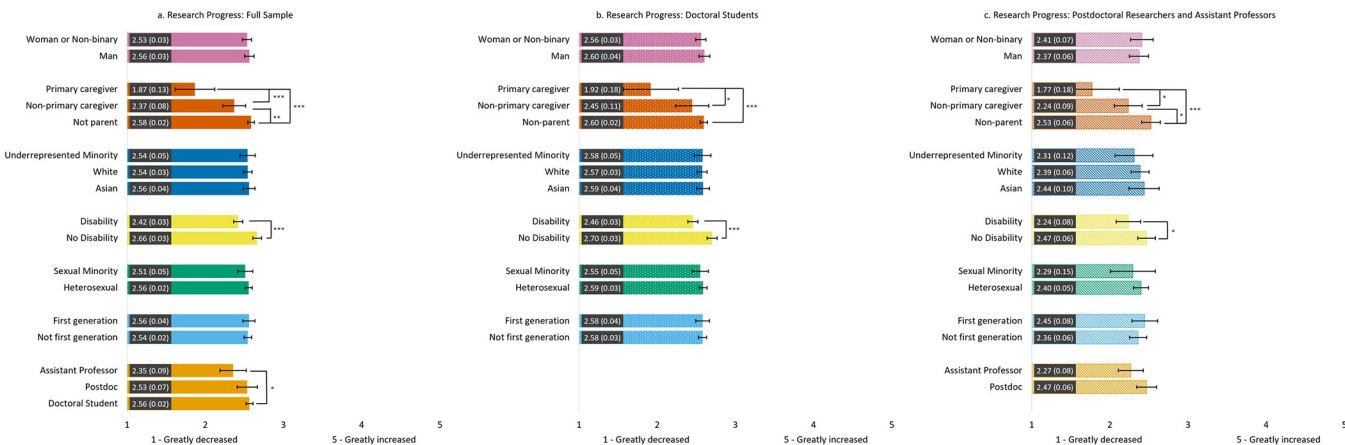

**Fig 1.** Estimated marginal means of reported change in research progress due to the COVID-19 pandemic for the full sample (1a) and the doctoral student (1b) and postdoc/asst prof (1c) subsamples. Participants were asked to rate the degree to which research progress changed on a Likert scale from 1 = *Greatly decreased* to 5 = *Greatly increased*. Error bars represent the 95% CI. Numeric values in each bar are the estimated marginal mean and standard error. * indicates p < .05, ** indicates p < .01, and *** indicates p < .001. See S3 Table in S1 File for results of each model.

sample, assistant professors reported a significant decrease in their research progress compared to doctoral students. However, differences in the impact of COVID-19 on research progress by gender, race, sexual identity, or first-generation status did not emerge across samples.

When examining subsamples, we found differences in the effect of parental caregiving on research progress. For all groups, parents who were primary caregivers reported a greater decrease in research progress than both non-primary caregivers and non-parents. However, for the full sample and the postdoc/asst prof subsample, non-primary caregivers also reported a significant decrease in research progress compared to non-parents.

**Change in workload (Fig 2).** Overall, all groups indicated that their workload had increased to some extent. We found that assistant professors reported a significantly greater increase in their workload compared to doctoral students and postdoctoral scholars. However,

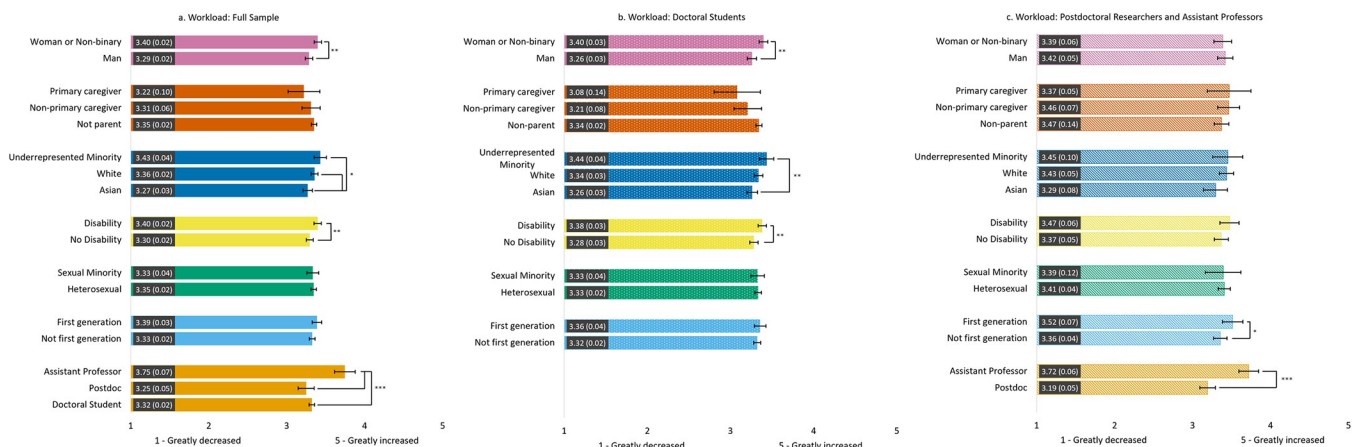

**Fig 2.** Estimated marginal means of reported change in workload due to the COVID-19 pandemic for the full sample (2a) and the doctoral student (2b) and postdoc/asst prof (2c) subsamples. Participants were asked to rate the degree to which their workload changed on a Likert scale from 1 = *Greatly decreased* to 5 = *Greatly increased*. Error bars represent the 95% CI. Numeric values in each bar are the estimated marginal mean and standard error. * indicates p < .05, ** indicates p < .01, and *** indicates p < .001. See S4 Table in S1 File for results of each model.

across the three samples, there were no differences in workload by parental caregiving or sexual identity.

When using the full sample or the doctoral subsample, we found that women and non-binary individuals and those with a disability reported a significantly greater increase in workload compared to men and those without a disability, respectively, but these differences did not emerge in the postdoc/asst prof subsample. When using the full sample, we found that URM and White individuals reported a significantly greater increase in their workload compared to Asian respondents. In the doctoral student subsample, URM individuals' workloads only differed from Asian individuals' workloads. No differences by race emerged in the postdoc/asst prof subsample. Finally, only in the postdoc/asst prof subsample, first-generation scholars reported a significantly greater increase in workload compared to non-first-generation scholars.

**Change in concern about career advancement (Fig 3).** All groups reported increased concern about their career advancement due to the COVID-19 pandemic. Across the three samples, those with a disability reported a significantly greater increase in concern about career advancement compared to those without a disability. Further, postdoctoral scholars reported a greater increase in these concerns compared to assistant professors and doctoral students. However, across samples, differences in COVID-19 related concerns about career advancement did not emerge based on gender, parental caregiving, sexual identity, or first-generation status.

We found subsample differences in concern about career advancement due to the pandemic by race. In the full sample, URM participants reported a greater increase in concern about their career advancement than White participants; in the doctoral student subsample, Asian participants reported greater increase in concern than White individuals; and for the postdoc/asst prof subsample, URM participants were significantly more concerned about career advancement than both White and Asian participants.

**Change in support from mentors (Fig 4).** Nearly all groups reported decreased mentor support, although means remained close to the scale midpoint. Across the three samples, those with a disability reported a significantly greater decrease in support from their mentors due to the pandemic compared to those without a disability. Further, assistant professors reported a significantly greater decrease in support from their mentors due to the pandemic compared to

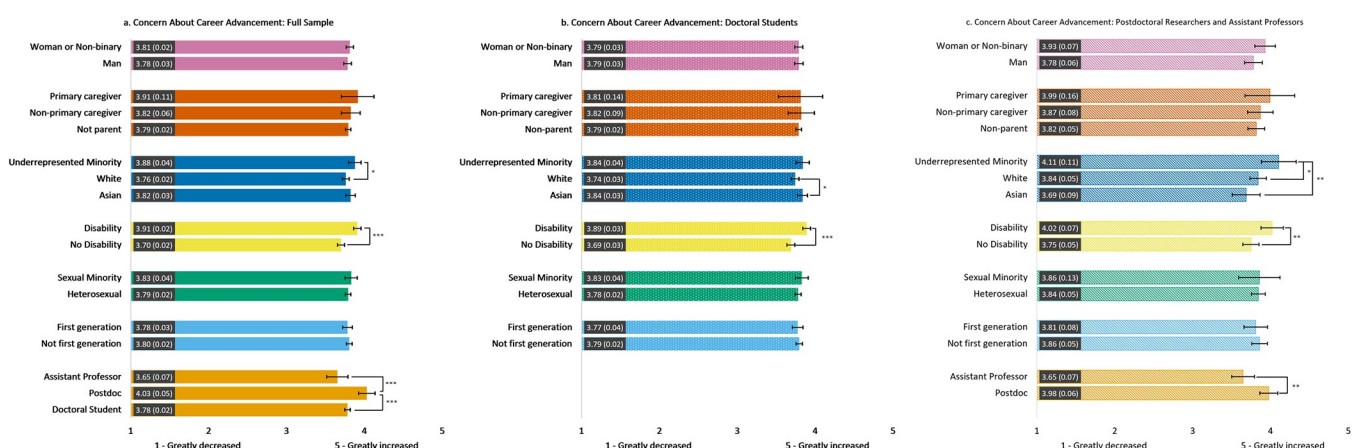

**Fig 3.** Estimated marginal means of reported change in concern about career advancement due to the COVID-19 pandemic for the full sample (3a) and doctoral student (3b) and postdoc/asst prof (3c) subsamples. Participants were asked to rate the degree to which their concern about career advancement changed on a Likert scale from 1 = *Greatly decreased* to 5 = *Greatly increased*. Error bars represent the 95% CI. Numeric values in each bar are the estimated marginal mean and standard error. * indicates p < .05, ** indicates p < .01, and *** indicates p < .001. See S5 Table in S1 File for results of each model.

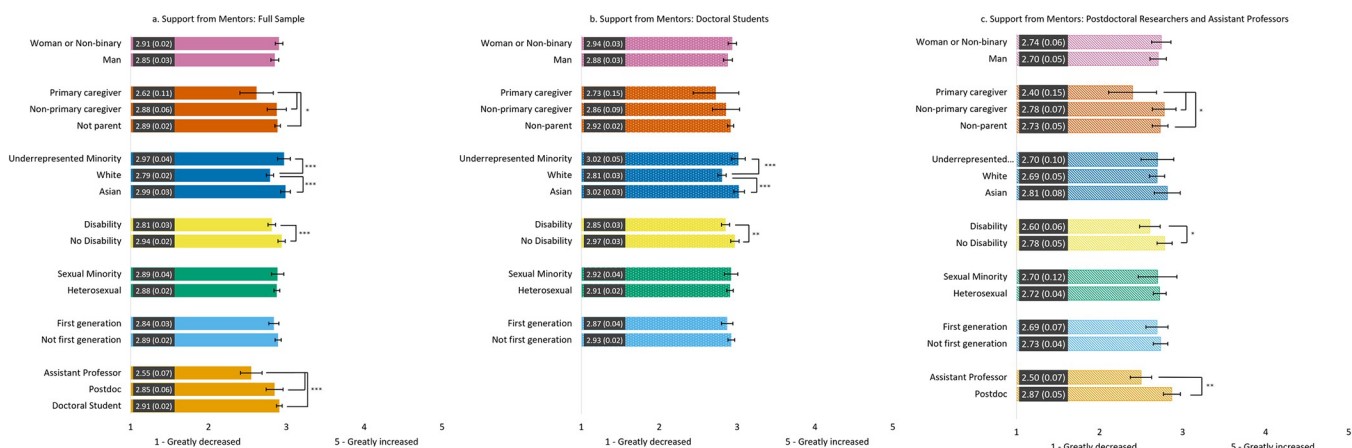

**Fig 4.** Estimated marginal means of reported change in mentor support due to the COVID-19 pandemic for the full sample (4a) and the doctoral student (4b) and postdoc/asst prof (4c) subsamples. Participants were asked to rate the degree to which their support from mentor(s) changed on a Likert scale from 1 = *Greatly decreased* to 5 = *Greatly increased*. Error bars represent the 95% CI. Numeric values in each bar are the estimated marginal mean and standard error. * indicates p < .05, ** indicates p < .01, and *** indicates p < .001. See S6 Table in S1 File for results of each model.

doctoral students and postdoctoral scholars. However, we did not find COVID-19 related differences in support from mentors by gender, sexual identity, or first-generation status.

We found subsample differences in support from mentors by parental caregiving and race. For the full sample and the postdoc/asst prof subsample, primary caregivers reported significantly greater decrease in mentor support than non-primary caregivers and non-parents; no differences were found among doctoral students. For the full sample and the doctoral subsample, we found that White participants reported a significantly greater decrease in support from their mentors due to the pandemic compared to both URM and Asian participants; no differences in mentoring support by race were found among the postdoc/asst prof subsample.

## Work disruptions due to COVID-19 life challenges

**Work disruptions due to physical health problems (Fig 5).** There were differences in the extent to which work was disrupted because of COVID-19 related physical health problems for many socio-demographic statuses. Across the three samples, women and non-binary individuals and those with a disability reported significantly greater work disruptions due to physical health problems compared to men and those without a disability, respectively. Further, doctoral students reported greater work disruptions due to physical health problems compared to postdoctoral scholars. Across samples, we did not observe differences in work disruptions due to physical health problems by sexual identity.

When examining the subsamples, we found several differences in work disruptions due to physical health problems. For the postdoc/asst prof subsample, we found that primary caregivers reported greater work disruptions due to physical health problems than non-primary caregivers and non-parents. For the full sample and the doctoral student subsample, we observed differences in physical health problems by race such that URM participants reported greater work disruptions than both White and Asian participants. Finally, for the full sample and doctoral student subsample, there were greater work disruptions due to physical health problems for first-generation scholars than non-first-generation scholars; these differences did not emerge in the postdoc/asst prof subsample.

**Work disruptions due to mental health problems (Fig 6).** Across the three samples, those with a disability and sexual minority participants reported a greater disruption to their

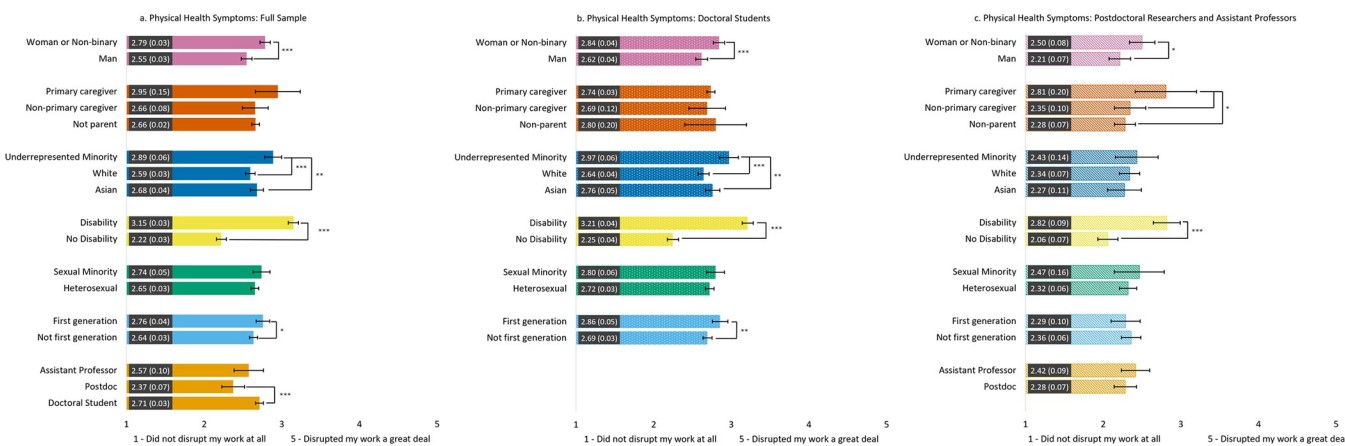

**Fig 5.** Estimated marginal means of disruptions to work due to physical health problems during the COVID-19 pandemic for the full sample (5a) and the doctoral student (5b) and postdoc/asst prof (5c) subsamples. Participants were asked to rate the degree to which physical health problems due to the COVID-19 pandemic disrupted their work with a Likert scale from 1 = *Did not disrupt my work* to 5 = *Greatly disrupted my work*. Error bars represent the 95% CI. Numeric values in each bar are the estimated marginal mean and standard error. * indicates $p < .05$, ** indicates $p < .01$, and *** indicates $p < .001$. See S7 Table in S1 File for results of each model.

work due to mental health problems compared to those without a disability and heterosexual individuals, respectively. Doctoral students reported a greater disruption to their work due to mental health problems compared to both postdoctoral scholars and assistant professors. Across samples, no differences in work disruptions due to mental health problems were found by first-generation status.

We found differences in work disruptions caused by mental health problems by gender, parental caregiving, and race. For the full sample and the postdoc/asst prof subsample, primary caregivers reported greater disruption to their work due to mental health problems compared to non-primary caregivers. In the full sample and doctoral student subsample, URM participants and women and non-binary participants reported more work disruptions due to mental health problems than Asian participants and men, respectively; these differences did not emerge in the postdoc/asst prof subsample.

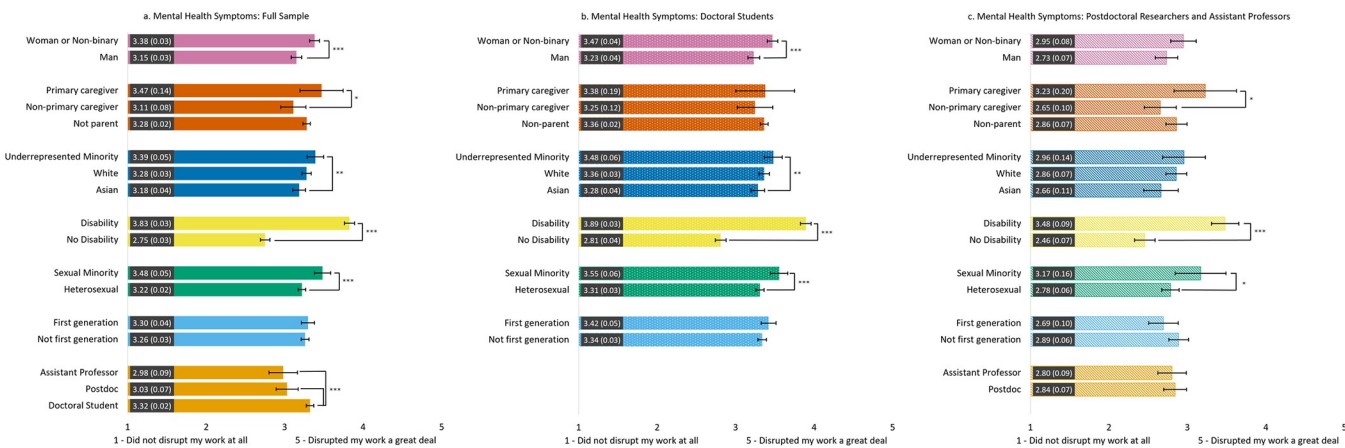

**Fig 6.** Estimated marginal means of disruptions to work due to mental health symptoms during the COVID-19 pandemic for the full sample (6a) and the doctoral student (6b) and postdoc/asst prof (6c) subsamples. Participants were asked to rate the degree to which mental health problems due to the COVID-19 pandemic disrupted their work with a Likert scale from 1 = *Did not disrupt my work* to 5 = *Greatly disrupted my work*. Error bars represent the 95% CI. Numeric values in each bar are the estimated marginal mean and standard error. * indicates $p < .05$, ** indicates $p < .01$, and *** indicates $p < .001$. See S8 Table in S1 File for results of each model.

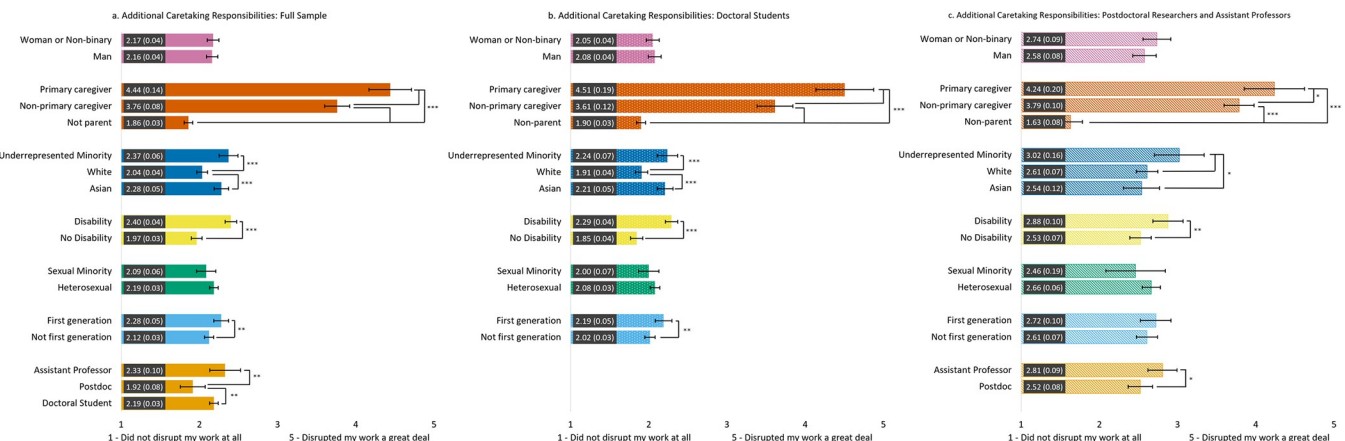

**Fig 7.** Estimated marginal means of disruptions to work due to additional caretaking responsibilities during the COVID-19 pandemic for the full sample (7a) and doctoral student (7b) and postdoc/asst prof (7c) subsamples. Participants were asked to rate the degree to which additional caretaking responsibilities due to the COVID-19 pandemic disrupted their work with a Likert scale from 1 = *Did not disrupt my work* to 5 = *Greatly disrupted my work*. Error bars represent the 95% CI. Numeric values in each bar are the estimated marginal mean and standard error. * indicates p < .05, ** indicates p < .01, and *** indicates p < .001. See supplemental Table S9 for results of each model.

**Work disruptions due to increased caregiving responsibilities (Fig 7).** Across the three samples, parents with primary caregiving responsibilities reported more work disruptions due to increased caregiving responsibilities than non-primary caregivers and non-parents; non-primary caregivers reported more work disruptions due to increased caregiving responsibilities than non-parents. Participants with a disability reported greater work disruptions due to additional caretaking responsibilities compared to those without a disability. Doctoral students and assistant professors also reported more work disruptions due to additional caretaking responsibilities than postdoctoral scholars. However, no differences in work disruptions due to increased caregiving were found by gender or sexual identity.

We found subsample differences in work disruptions due to additional caretaking by race and first-generation status. For both the full sample and the doctoral subsample, URM and Asian participants reported greater work disruptions due to additional caretaking responsibilities compared to White participants; however, for the postdoc/asst prof subsample, URM participants reported greater work disruptions due to additional caretaking responsibilities compared to White and Asian participants. When using the full sample and the doctoral student subsample, we found that first-generation scholars reported more work disruptions due to additional caretaking than non-first-generation participants.

## Discussion

Our results (summarized in Tables 1 and 2) indicate a wide array of consequences of living and working through the COVID-19 pandemic for early-career scientists. These consequences are not evenly shouldered across academic scholars. Consistent with our hypothesis, individuals from marginalized groups reported more work disruptions due to COVID-19 than those from more privileged groups. Further, important differences were observed among scholars with different socio-demographic statuses. Changes in work outcomes (e.g., research progress) most often differed by parental caregiving, race, disability status, and career stage, whereas work disruptions due to COVID-19 related life challenges (e.g., physical health problems) were seen across most socio-demographic groups. We describe these patterns below and offer recommendations for support mechanisms for marginalized early-career scientists.

**Table 1. Summary of results for all socio-demographic statuses and change in COVID-19 work impacts.**

| | Change (*greatly decreased* to *greatly increased*) in: | | | |
| --- | --- | --- | --- | --- |
| | **Research progress** | **Workload** | **Concern about career[1]** | **Support from mentor(s)** |
| Gender: • Full Sample | | Women/Non-bin.>Men | | |
| • Doc. Sample | | Women/Non-bin.>Men | | |
| • PD/AP Sample | | | | |
| Parental caregiving: • Full Sample | Primary < Non-primary < Non-parent | | | Primary < Non-primary & Non-parent |
| • Doc. Sample | Primary < Non-prim. & Non-parent | | | |
| • PD/AP Sample | Primary < Non-primary < Non-parent | | | Primary < Non-primary & Non-parent |
| Race: • Full Sample | | URM & White > Asian | URM > White | White < URM & Asian |
| • Doc. Sample | | URM > Asian | Asian > White | White < URM & Asian |
| • PD/AP Sample | | | URM > White & Asian | |
| Disability: • Full Sample | Disability < No dis. | Disability > No dis. | Disability > No dis. | Disability < No dis. |
| • Doc. Sample | Disability < No dis. | Disability > No dis. | Disability > No dis. | Disability < No dis. |
| • PD/AP Sample | Disability < No dis. | | Disability > No dis. | Disability < No dis. |
| Sexual identity[2] | | | | |
| First-gen. • Full Sample | | | | |
| • Doc. Sample | | | | |
| • PD/AP Sample | | First-gen. > Not first-gen. | | |
| Career stage: • Full Sample | Asst. prof. < Doc. | Asst. prof. > Doc. & Postdoc | Postdoc > Doc. & Asst. prof. | Asst. prof. < Doc. & Postdoc |

*Note*. Grey shaded squares indicate no differences by status. Abbreviations used in the table are defined as follows: 'Full Sample' is all participants in the sample; 'Doc. Sample' is subsample of doctoral students; 'PD/AP Sample' is subsample of postdoctoral researchers and doctoral students; 'Non-bin.' is non-binary; 'Primary' is primary parental caregiver; 'Non-primary' is non-primary parental caregiver; 'URM' is underrepresented minority; 'Hetero' is heterosexual; 'First-gen.' is first-generation college student; 'Doc. Student' is doctoral student; 'Postdoc' is postdoctoral scholar; 'Asst. prof.' is assistant professor.

[1] Concern about career advancement.

[2] No differences in any sample for sexual identity status.

The pandemic's impact was particularly notable for early-career scientists with disabilities. Specifically, those who reported a physical or mental disability were significantly more likely to experience all seven types of work disruptions examined in the study. There has been limited systematic research on the experiences of scientists with at least one disability [22]. However, our results highlight the importance of understanding their experiences. Many scholars with disabilities were not provided with appropriate accommodations prior to the pandemic, and have often been left out of the narrative about shifting to work-from-home and remote teaching throughout the COVID-19 pandemic [22]. Our findings further support existing discussions of the need for mental health resources, mentorship, and material support to increase the inclusion of academics with mental and physical disabilities [23].

Generally, URM participants reported more negative changes in work outcomes than other racial groups, extending prior findings of increased COVID-related stress and burnout among faculty of color [3, 16–19]. URM scholars in our sample, compared to other racial groups, reported a greater increase in workload, growing concern about career advancement, and more work disruptions from COVID-related physical health problems, mental health problems, and increased caretaking responsibilities. However, in the full sample White participants reported the greatest decrease in mentor support and, similar to URM scholars, White

**Table 2. Summary of results for all socio-demographic statuses and COVID-19 work disruptions due to life challenges.**

| | Work disrupted (*not at all* to *greatly*) due to: | | |
| --- | --- | --- | --- |
| | **Physical health symptoms** | **Mental health symptoms** | **Additional caretaking[1]** |
| Gender:<br>• Full Sample | Women/Non-bin.>Men | Women/Non-bin.>Men | |
| • Doc. Sample | Women/Non-bin.>Men | Women/Non-bin.>Men | |
| • PD/AP Sample | Women/Non-bin.>Men | | |
| Parental caregiving:<br>• Full Sample | | Primary > Non-primary | Primary > Non-primary > Non-parent |
| • Doc. Sample | | | Primary > Non-primary > Non-parent |
| • PD/AP Sample | Primary > Non-primary & Non-parent | Primary > Non-primary | Primary > Non-primary > Non-parent |
| Race[2]:<br>• Full Sample | URM > White & Asian | URM > Asian | URM & Asian > White |
| • Doc. Sample | URM > White & Asian | URM > Asian | URM & Asian > White |
| • PD/AP Sample | | | URM > Asian & White |
| Disability:<br>• Full Sample | Disability > No disability | Disability > No disability | Disability > No disability |
| • Doc. Sample | Disability > No disability | Disability > No disability | Disability > No disability |
| • PD/AP Sample | Disability > No disability | Disability > No disability | Disability > No disability |
| Sexual identity:<br>• Full Sample | | Sexual minority > Hetero. | |
| • Doc. Sample | | Sexual minority > Hetero. | |
| • PD/AP Sample | | Sexual minority > Hetero. | |
| First-gen.:<br>• Full Sample | First-gen. > Not first-gen. | | First-gen. > Not first-gen. |
| • Doc. Sample | First-gen. > Not first-gen. | | First-gen. > Not first-gen. |
| • PD/AP Sample | | | |
| Career stage:<br>• Full Sample | Doc. student > Postdoc | Doc. student > Postdoc & Asst. prof. | Doc. student & Asst. prof. > Postdoc |

*Note*. Grey shaded squares indicate no differences by status. Abbreviations used in the table are defined as follows: 'Full Sample' is all participants in the sample; 'Doc. Sample' is subsample of doctoral students; 'PD/AP Sample' is subsample of postdoctoral researchers and doctoral students; 'Non-bin.' is non-binary; 'Primary' is primary parental caregiver; 'Non-primary' is non-primary parental caregiver; 'URM' is underrepresented minority; 'Hetero' is heterosexual; 'First-gen.' is first-generation college student; 'Doc. Student' is doctoral student; 'Postdoc' is postdoctoral scholar; 'Asst. prof.' is assistant professor.

[1] Additional caretaking responsibilities at home.

participants reported a greater increase in their workload compared to Asians. We postulate that mentors with more limited capacity to provide support because of the pandemic may have directed that support towards mentees of color (URM and Asian) rather than White mentees.

Our findings about the effects of COVID-19 on Asian participants' work outcomes were sometimes more similar to URM participants and sometimes more similar to White participants. For example, similar to URM doctoral students, Asian doctoral students reported more work disruptions due to additional caretaking responsibilities than White students, and less of a decrease in support from mentors. However, similar to White doctoral students, Asian students reported fewer work disruptions due to physical health problems than URM scholars. The complexity of the findings for Asian scholars may reflect that in STEM, this group can experience marginalization even when they are not underrepresented and suggests that more research focused on their COVID-related experiences is warranted. More nuanced work on Asian scholars' COVID experiences in academia is particularly important given the increase in

anti-Asian racism and violence that grew in the United States in 2020 and 2021 in part due to the COVID-19 pandemic [24, 25].

Furthermore, we found some subsample differences in COVID-19 outcomes related to race. Most often, we observed similar findings in the overall sample and the doctoral student subsample that did not emerge in the postdoc/asst prof subsample; this was the case for workload, mentor support, and work disruptions from both physical health and mental health symptoms. These differences may be due to a lack of power to detect differences due to the smaller numbers of Asian and URM participants. However, in some cases, the pattern of means differed across subsamples, suggesting that the effect of racial marginalization may differ at the doctoral stage as compared to later career stages, perhaps due to career-stage changes in demographic representation.

A few differences in work outcomes emerged by sexual identity and first-generation status. In particular, first-generation college students who were postdocs or assistant professors reported a greater increase in their workload compared to non-first-generation scholars at the same career stage. Further, although no other differences emerged in work impacts by first-generation status (e.g., research progress), first-generation doctoral students experienced more work disruptions due to physical health problems and additional caregiving responsibilities. These findings suggest that more research on first-generation scholars is warranted in order to better understand the impact of COVID-19 on their careers over time. Sexual minority participants reported more disruptions due to mental health problems. This last result is consistent with a recent study of LGBT+ college students that found nearly 60% of the students sampled experienced increased anxiety and distress during the COVID-19 pandemic, notably associated with lower levels of family support [26].

Notably, our findings that women and gender non-binary participants reported greater work disruptions due to physical and mental health problems than men are consistent with prior research findings on the greater stress and burnout reported by women during the pandemic [27]. However, although early research found strong gender differences in productivity and other COVID-19 work impacts [3, 9–12], our results showed less consistent patterns in these areas. We found that compared to men, women and gender non-binary doctoral students reported a greater increase in their workload due to COVID-19, but no differences emerged in research progress, concern about career advancement, or support from mentors. We may have found fewer gender differences in work disruptions than other studies in the literature because we examined differences in subjective perceptions of COVID-19 impacts (e.g., research progress) as opposed to objective measures (e.g., number of publications). In making their subjective assessments of impacts like research progress, participants may have compared themselves to others in their group (e.g., women compared themselves to other women and men to other men) facing similar types of COVID-19-related challenges. This comparison might have led women and gender non-binary individuals to minimize the effect of COVID-19 on their work.

We found that participants with more parental caregiving demands reported significantly greater decreases in research progress and greater work disruptions that they attributed to additional caregiving responsibilities. For the postdoc/asst prof subsample, these greater caregiving demands were also associated with less support from mentors, and greater work disruptions due to physical and mental health problems. These results underscore findings from other research indicating that it has been challenging for parents to navigate workplace responsibilities during the pandemic [21]. Differences by parental caregiving were more likely to emerge for the postdoc/asst prof subsample than the doctoral subsample, likely because our sample contained a greater percentage of assistant professors who were parents (54%) than postdocs (21%) or doctoral students (6%) who were parents.

Our research extends previous work documenting the challenges faced by assistant professors during the pandemic by comparing this group to other early-career scholars (i.e., doctoral students and postdoctoral scholars), finding differences across all seven work outcomes examined. Compared to doctoral students and/or postdocs, assistant professors reported the greatest decrease in their research progress and support from mentors and the largest increases in workload and work disruptions due to additional caretaking responsibilities. These results extend previous research that demonstrated disproportionately high impacts of COVID-19 on early-career faculty compared to mid- and later-career faculty members which were theorized to reflect a "perfect storm" of demands on tenure-track assistant professors–pressure for productivity to achieve tenure, the need for mentoring to support them in a new career stage, and the likelihood of having young children [27].

Work disruptions due to physical and mental health problems during the pandemic were particularly acute for doctoral students, likely aggravating the mental health crisis among graduate students that existed before the pandemic [28]. Although postdoctoral scholars reported lesser increases in workload and work disruptions due to caretaking responsibilities than other groups, postdocs were the most concerned about their career advancement, perhaps due to pandemic-related hiring freezes and postponements. Taken together, these results point to the likely long-term negative effects of COVID-19 on academic science, particularly if these work disruptions are related to career intentions.

## Limitations

The data presented in this paper were part of a study about early career experiences in academia. Although an important dataset, we recognize some specific limitations. First, given the time of data collection (April-May of 2021), validated measures of COVID-19 impacts had not been developed. Therefore, we included only single-item measures of COVID-19 impacts on research and academic work. Second, we measured respondents' *subjective* experiences rather than objective measures such as the number of manuscripts submitted to journals [1–3]. However, measures of subjective experiences at work and school are critical, as prior literature indicates that they strongly predict organizational satisfaction and intentions to remain on a science career path [29, 30].

Third, there are limitations related to the relatively small number of participants within particular career stages or socio-demographic groups. Although response rates were robust, our small sample of postdoctoral scholars and assistant professors required us to pool these two career stages in the subsample analyses rather than analyze them separately. Even combined, we may not have had enough power to detect some small but important effects in this subsample. Further, although we capture differences by socio-demographic status such as race and gender, our findings do not speak to the unique intersectional impacts of COVID-19 for those occupying multiple marginalized and minoritized statuses, such as Black women or sexual minority caregivers, for whom multiple systems of inequality may create additional and unique demands and challenges [31].

Finally, our project examines scholars' experiences in four fields that span the spectrum of life, physical, and social sciences (biology, economics, physics, and psychology). Although our data do not speak directly to the outcomes of the COVID-19 pandemic beyond these fields, other research has identified that scholars in the arts and humanities have faced significant, long-term barriers to their scholarly work due to the pandemic [3].

## Implications and recommendations

An understanding of the breadth of COVID-19 impacts and their differential consequences for scholars across socio-demographically marginalized communities is critical for university

administrators and others to determine the best way to respond equitably to pandemic-related disruptions. Our results improve understanding of how inequalities are reproduced in the academy and point to career stages and socio-demographic groups that may require additional support to recover from the pandemic professionally. Our findings suggest the need for longitudinal studies that track the long-term impacts of COVID-19 on career outcomes of early-career scholars, particularly for scholars from marginalized groups. Our supplemental analyses (see S12-S16 Tables in S1 File) indicated that COVID-19 disruptions were associated with negative work outcomes; thus, the socio-demographic differences in COVID-19 disruptions that we observed may result in reduced job satisfaction and professional role confidence while increasing turnover intentions and burnout for minoritized and marginalized early-career academics. Without data-driven policies and structural supports, academic science risks a significant and long-term loss of diversity.

To limit the loss of early-career scholars from marginalized and minoritized groups, it is crucial that policies be adopted that provide institutional support to those affected most significantly by the pandemic. Although the acute challenges of the pandemic will likely diminish over the coming years, COVID-19 impacts may be a sort of "canary in the coal mine," highlighting areas where social crises (e.g., COVID-19 pandemic, racial injustice, #MeTooPhD) amplify already existing inequalities for academics from marginalized groups. For example, the greater work disruption due to physical health problems we found for women and non-binary individuals, primary caregiver postdoctoral scholars and assistant professors, URM individuals, and those with disabilities, may indicate that the academic workload was already disproportionally distributed, with more of the burden–likely including invisible service and other unrecognized labor–being carried by individuals in these groups [32]. Thus, the following recommendations are both specific to addressing the negative impact of COVID-19 and broadly aimed at creating a more inclusive academic science context.

The pandemic has illustrated that certain practices, such as remote options for meetings, conferences, and professional development opportunities, increase accessibility for individuals with disabilities and/or caregiving responsibilities. Similarly, recording talks or webinars creates flexibility for those balancing many competing responsibilities. A study of disabled workers prior to the pandemic found that only 5% of respondents were able to work from home, despite remote work often being preferable for individuals with disabilities [33]. The positive impact of certain remote working practices challenges academia to reconsider a "return-to-normal," and instead create a more inclusive working environment by maintaining multi-modal forms of participation in conferences, talks, and networking events for early-career researchers, as well as opportunities for remote and hybrid work.

Some of the findings of our research pointed to the perceived loss of mentoring support concurrent with an increase in mental and physical health problems for early-career scientists. Therefore, providing additional mentor support during the next few years, especially for assistant professors who have lost much of this support during the pandemic, could be an important step for ensuring their success in academia. At the same time, it is also important to recognize the range of other service commitments placed on mentors, especially those from underrepresented groups, and provide them with the time and resources needed to support early-career scholars. Additionally, institutions need to dramatically increase their investment in mental and physical health resources, particularly for the support of doctoral students, gender and sexual minority scholars, primary caregivers, and persons with disabilities.

Our findings of reduced research progress and accompanying career-related concerns suggest the need for institutions to account for pandemic-related work challenges in institutional rewards systems. Universities can provide resources, including research funds, along with opportunities for faculty, postdoctoral scholars, and doctoral students to document the

impacts of the pandemic on their work so that steps can be taken to alleviate those impacts [34]. Moreover, institutions will need to thoughtfully consider how to adjust their evaluations (e.g., annual reviews, promotion and tenure, and hiring processes) to account for differences in academic records due to COVID-19's disparate impact on marginalized communities [6, 8, 35]. These include holistic review practices that weigh quality over quantity and review criteria that account for the visible and invisible service and emotional labor that is often performed by scholars from marginalized groups. These steps are urgently needed to maintain the future strength and diversity of the STEM workforce.

## Supporting information

**S1 File.**
(DOCX)

## Acknowledgments

The authors would like to thank Dr. Petal Grower for her feedback and revisions on this manuscript as well as her contributions to the CLIMBS UP team. We would also like to thank Alexander D. Garcia-Settles for his assistance preparing tables for the manuscript.

## Author Contributions

**Conceptualization:** Isis H. Settles, Erin A. Cech, Georgina M. Montgomery, Kevin C. Elliott, Kendra Spence Cheruvelil.

**Data curation:** Hannah M. Douglas, Isis H. Settles, Erin A. Cech, Guizhen Ma, Tangier M. Davis.

**Formal analysis:** Hannah M. Douglas, Isis H. Settles, Erin A. Cech.

**Funding acquisition:** Isis H. Settles, Erin A. Cech, Georgina M. Montgomery, Kevin C. Elliott, Kendra Spence Cheruvelil.

**Investigation:** Hannah M. Douglas, Isis H. Settles, Erin A. Cech, Guizhen Ma, Tangier M. Davis.

**Methodology:** Hannah M. Douglas, Isis H. Settles, Erin A. Cech, Georgina M. Montgomery, Guizhen Ma, Tangier M. Davis, Kevin C. Elliott, Kendra Spence Cheruvelil.

**Project administration:** Hannah M. Douglas, Isis H. Settles, Guizhen Ma, Kendra Spence Cheruvelil.

**Supervision:** Isis H. Settles, Kendra Spence Cheruvelil.

**Visualization:** Hannah M. Douglas, Isis H. Settles, Lexi R. Nadolsky, Arika K. Hawkins.

**Writing – original draft:** Hannah M. Douglas, Isis H. Settles, Erin A. Cech, Georgina M. Montgomery, Kevin C. Elliott, Kendra Spence Cheruvelil.

**Writing – review & editing:** Hannah M. Douglas, Isis H. Settles, Erin A. Cech, Georgina M. Montgomery, Lexi R. Nadolsky, Arika K. Hawkins, Guizhen Ma, Tangier M. Davis, Kevin C. Elliott, Kendra Spence Cheruvelil.

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
