## [Decision Letter · Decision Letter 0]

26 Apr 2022

PONE-D-22-07647Disproportionate impacts of COVID-19 on marginalized and minoritized early-career academic scientistsPLOS ONE

Dear Dr. Douglas,

Thank you for submitting your manuscript to PLOS ONE. After careful consideration, we feel that it has merit but does not fully meet PLOS ONE’s publication criteria as it currently stands. Therefore, we invite you to submit a revised version of the manuscript that addresses the points raised during the review process.

The reviewers felt this was an important piece of work that can inform policy development in higher education to mitigate the inequitable effects of the COVID-19 pandemic. Reviewer 2 offers an important critique of the analysis and reporting of the findings from postdoc and faculty sub-samples, given that they were much smaller than that of graduate students, which I ask you to particularly address. I look forward to receiving a revised version of this work.

We look forward to receiving your revised manuscript.

Kind regards,

Quinn Grundy, PhD, RN

Academic Editor

PLOS ONE

Journal Requirements:

Reviewers' comments:

Reviewer's Responses to Questions

**Comments to the Author**

1. Is the manuscript technically sound, and do the data support the conclusions?

Reviewer #1: Yes

Reviewer #2: Partly

2. Has the statistical analysis been performed appropriately and rigorously? 

Reviewer #1: Yes

Reviewer #2: No

3. Have the authors made all data underlying the findings in their manuscript fully available?

Reviewer #1: Yes

Reviewer #2: No

4. Is the manuscript presented in an intelligible fashion and written in standard English?

Reviewer #1: Yes

Reviewer #2: Yes

5. Review Comments to the Author

Reviewer #1: This is a well-executed and well-written assessment of the varied impact of the COVID-19 pandemic on early-career individuals in STEM. The sample sizes are adequate for most groups except for the small number of individuals who self-identify as non-binary. However, the authors note that these individuals tended to follow similar trends as women and were able to combine them with this group. While these findings are unable to describe the intersectionality will which often exacerbate many of these socio-demographic differences, this work is a great first step in understanding how this pandemic will disproportionately impact those groups that are already historically marginalized or excluded from the STEM fields and academia. These data are necessary for university to enact policy that specifically aids these groups in recovery from this pandemic in order to not further advance the minimization of these groups in STEM/academia. I strongly encourage publication of this work.

Reviewer #2: This is a timely study examining the effects of COVID on the academic progress of early career scientists. While in general I think the survey methods are clear the authors do need to provide more information concerning when and how the survey was distributed as well as including a copy of the survey itself in the supplemental. Otherwise the data are not reproducible. I also do have some major concerns about the statistical analyses. Based on Supplemental Tables 1 and 2, the authors do not have enough faculty or postdoc respondents to examine the effects of race, gender, disability, or parental status on response to COVID for those two career stages. Given the high number of graduate student respondents, this paper would be much stronger if the authors only used the graduate student data for all responses except for the comparisons across career stage. Since assistant professors make up less than 10% of the responses, and postdocs make up less than 20% of the responses, they have no impact on the results for broad demographic groups. The reported results are really just the results for graduate student participants.

This overwhelming difference in number of responses from graduate students relative to postdocs and faculty has downstream effects on the validity of the authors’ interpretation of their results, some of which I mention in my line comments below.

Line Comments:

Line 40: “other work dimensions” is vague. Please be more specific.

Lines 56-63: My understanding is that most of these studies have focused on faculty, not graduate students or postdocs. Since the authors specifically query graduate students and postdocs in this study, I suggest the authors mention the career stage targeted by the studies mentioned here, as individuals at different career stages may have different experiences.

An additional, more minor point here is that these cited studies primarily focus on effects of caregiving. There are virtually no studies examining effects of gender on response to COVID in the absence of caregiving. In other words, there are no studies comparing the effects of COVID on single women, men, and non-binary individuals without children in academia. This is an important distinction, because “woman” and “caregiver” are often conflated. Since the authors attempt to tease this apart in their own work, it is worth pointing out that the previous studies have not.

Line 99: I suggest including the specific year this study was conducted instead of stating “the past year”, as it will be read in future years. Please also include the months this survey was live.

Since this is largely a study of graduate students (10x as many graduate student respondents as professors) I suggest the authors emphasize their survey of graduate students.

Line 112: What do the authors mean by “prestige”? Please define or use a different term. Based on the supplemental material, it looks like they mean R1, R2, R3. This is not a definition of prestige, this is instead Carnegie Classification of Institutes of Higher Education.

Line 114: Please include IRB protocol numbers

Line 120: Please include justification for aggregating non-binary with woman.

For Figures 1 and 2: I suggest the authors put the sample size per bar above/next to each bar. Otherwise the visualization is a false equivalency because it implies even sample sizes across categories.

Lines 281-290: This result could be largely due to the differential experiences of female graduate students, postdocs, and faculty, since the respondents are overwhelmingly graduate students. This illustrates my earlier point that it is extremely important to address the fact that faculty and postdoc respondents only make up just over 20% of the total respondents, so their voices are not heard in any of the analyses that analyze graduate students, postdocs, and faculty in aggregate.

Line 304: This statement would be stronger if the authors compared the proportion of respondents that have children by career stage (% grad students vs % postdocs vs % assistant professors)

Line 307: I suggest removing the quotes around “mental health crisis” here, as that might indicate to readers that the authors don’t believe graduate students are having a mental health crisis (which I don’t think is case).

Line 319: Here the authors comment on the timing of their survey, but since they did not include the specific dates of the survey in their methods section this statement is ungrounded. Please add the year and months this survey was live in the methods, and re-state the time frame of the survey here, so the readers do not need to flip back to the methods section to find it.

Lines 335: Please include the small sample size of assistant professors and postdocs, as well as indicating why the study did not speak to intersectionality (low sample size for each category).

Supplemental Material, Participant Database: Please include the number of respondents per group in addition to the number of individuals queried.

Please include consistent line numbers throughout so reviewers don’t have to do math to make line comments.

6. PLOS authors have the option to publish the peer review history of their article (what does this mean?). If published, this will include your full peer review and any attached files.

Reviewer #1: No

Reviewer #2: No

---

## [Author Response · Author response to Decision Letter 0]

1 Jul 2022

Reviewer #1:

This is a well-executed and well-written assessment of the varied impact of the COVID-19 pandemic on early-career individuals in STEM. The sample sizes are adequate for most groups except for the small number of individuals who self-identify as non-binary. However, the authors note that these individuals tended to follow similar trends as women and were able to combine them with this group. While these findings are unable to describe the intersectionality will which often exacerbate many of these socio-demographic differences, this work is a great first step in understanding how this pandemic will disproportionately impact those groups that are already historically marginalized or excluded from the STEM fields and academia. These data are necessary for university to enact policy that specifically aids these groups in recovery from this pandemic in order to not further advance the minimization of these groups in STEM/academia. I strongly encourage publication of this work.

Thank you for your review and comments of our work!

Reviewer #2:

This is a timely study examining the effects of COVID on the academic progress of early career scientists. While in general I think the survey methods are clear the authors do need to provide more information concerning when and how the survey was distributed as well as including a copy of the survey itself in the supplemental. Otherwise the data are not reproducible. I also do have some major concerns about the statistical analyses. Based on Supplemental Tables 1 and 2, the authors do not have enough faculty or postdoc respondents to examine the effects of race, gender, disability, or parental status on response to COVID for those two career stages. Given the high number of graduate student respondents, this paper would be much stronger if the authors only used the graduate student data for all responses except for the comparisons across career stage. Since assistant professors make up less than 10% of the responses, and postdocs make up less than 20% of the responses, they have no impact on the results for broad demographic groups. The reported results are really just the results for graduate student participants.

Thank you for your review of our manuscript. To address the main concern regarding the relatively small sample size of postdocs and assistant professor compared to doctoral students, we replicated the 7 multiple regressions on two different subsamples (7 models exploring COVID-19 impacts for doctoral students alone and 7 models analyzing COVID-19 impacts for postdocs and assistant professors combined). In the results we present the consistent findings across all models as well as describe where results differ across subsamples. These analyses demonstrated that many differences were visible across subsamples, whereas others emerged only for some subsamples. Notably, there were cases where findings emerged only for the postdoc/assistant professor subsample despite their smaller numbers. As our overall findings were not simply a reflection of graduate students’ experiences, we choose to retain our focus on early-career researchers generally, rather than only focusing on doctoral students. This provides a more holistic understanding of negative effects of COVID-19 on academics across early career stages. With the updated presentation of results and discussion in our resubmission, we are able to show some of the universal experiences of this pandemic while also highlighting unique impacts across career stages.

This overwhelming difference in number of responses from graduate students relative to postdocs and faculty has downstream effects on the validity of the authors’ interpretation of their results, some of which I mention in my line comments below.

Line Comments:

Line 40: “other work dimensions” is vague. Please be more specific.

We’ve specified other work dimensions as increased workload and decreased research progress (LINE 50).

Lines 56-63: My understanding is that most of these studies have focused on faculty, not graduate students or postdocs. Since the authors specifically query graduate students and postdocs in this study, I suggest the authors mention the career stage targeted by the studies mentioned here, as individuals at different career stages may have different experiences.

We have added the career stage of the studies cited in the introduction. When the cited studies surveyed people in academia but did not specify career stage, we referred to the sample as “academics.” (LINES 69 – 72)

An additional, more minor point here is that these cited studies primarily focus on effects of caregiving. There are virtually no studies examining effects of gender on response to COVID in the absence of caregiving. In other words, there are no studies comparing the effects of COVID on single women, men, and non-binary individuals without children in academia. This is an important distinction, because “woman” and “caregiver” are often conflated. Since the authors attempt to tease this apart in their own work, it is worth pointing out that the previous studies have not.

Thank you for this observation about the state of the research. In our introduction, we have cited studies that do look at gender and caregiving separately and jointly (e.g., Myers et al. - Nature Human Behavior; Deryugina et al, 2021).

Line 99: I suggest including the specific year this study was conducted instead of stating “the past year”, as it will be read in future years. Please also include the months this survey was live.

We’ve added the time frame and the months the survey was open (LINES 111 AND 119).

Since this is largely a study of graduate students (10x as many graduate student respondents as professors) I suggest the authors emphasize their survey of graduate students.

Since we have included the additional analyses by subgroup, we have emphasized the differences and similarities across career stages throughout the document.

Line 112: What do the authors mean by “prestige”? Please define or use a different term. Based on the supplemental material, it looks like they mean R1, R2, R3. This is not a definition of prestige, this is instead Carnegie Classification of Institutes of Higher Education.

We defined prestige by the National Research Council rankings of PhD granting institutions; we added this detail to the main text (LINE 159). First, we got the ranked list of universities for each of the four departments and divided them into terciles (our ranking definition). We randomly selected 10 universities from each tercile. This more detailed description of the prestige variable is found in the supplement (LINES 5-8).

Line 114: Please include IRB protocol numbers

The IRB protocol numbers have been added to the method (LINE 126).

Line 120: Please include justification for aggregating non-binary with woman.

The justification for aggregating these groups was added to the method (LINES 143-148).

For Figures 1 and 2: I suggest the authors put the sample size per bar above/next to each bar. Otherwise the visualization is a false equivalency because it implies even sample sizes across categories.

Thank you for this suggestion, we agree that the subgroup sample size is important. We decided to add that information to the summary of results tables instead of the figures so as to retain the readability of the figures (which already contain the mean and standard error). 

Lines 281-290: This result could be largely due to the differential experiences of female graduate students, postdocs, and faculty, since the respondents are overwhelmingly graduate students. This illustrates my earlier point that it is extremely important to address the fact that faculty and postdoc respondents only make up just over 20% of the total respondents, so their voices are not heard in any of the analyses that analyze graduate students, postdocs, and faculty in aggregate.

After re-running the analyses in the postdoc and assistant professor sample, we found that advanced early-career women did not significantly differ from men in any of the work impacts where there weren’t also differences in the doctoral student sample.

Line 304: This statement would be stronger if the authors compared the proportion of respondents that have children by career stage (% grad students vs % postdocs vs % assistant professors)

Thank you for this suggestion; we’ve added those details (LINE 469-470).

Line 307: I suggest removing the quotes around “mental health crisis” here, as that might indicate to readers that the authors don’t believe graduate students are having a mental health crisis (which I don’t think is case).

Thank you for this suggestion; we’ve removed the quotes.

Line 319: Here the authors comment on the timing of their survey, but since they did not include the specific dates of the survey in their methods section this statement is ungrounded. Please add the year and months this survey was live in the methods, and re-state the time frame of the survey here, so the readers do not need to flip back to the methods section to find it.

This has been added to the limitations section (LINE 495).

Lines 335: Please include the small sample size of assistant professors and postdocs, as well as indicating why the study did not speak to intersectionality (low sample size for each category).

The smaller sample size for assistant professors and postdocs has been added to the limitation section (LINE 503). We do speak to the lack of an intersectional lens in the limitations, though it is not necessarily due to sample size, but a function of our analytic plan (508 - 512).

Supplemental Material, Participant Database: Please include the number of respondents per group in addition to the number of individuals queried.

We added the number of participants per group in the supplemental tables S3-S9.

Please include consistent line numbers throughout so reviewers don’t have to do math to make line comments.

Sorry for that! The line number formatting is updated.

---

## [Decision Letter · Decision Letter 1]

9 Aug 2022

PONE-D-22-07647R1Disproportionate impacts of COVID-19 on marginalized and minoritized early-career academic scientistsPLOS ONE

Dear Dr. Montgomery,

Thank you for submitting your manuscript to PLOS ONE. After careful consideration, we feel that it has merit but does not fully meet PLOS ONE’s publication criteria as it currently stands. Therefore, we invite you to submit a revised version of the manuscript that addresses the points raised during the review process.

Thank you for your thoughtful and comprehensive response to the previous reviews. The manuscript is much strengthened. The reviewer has a few minor, outstanding comments for your consideration, which I ask you to address. I look forward to receiving your revised manuscript.

We look forward to receiving your revised manuscript.

Kind regards,

Quinn Grundy, PhD, RN

Academic Editor

PLOS ONE

Journal Requirements:

Reviewers' comments:

Reviewer's Responses to Questions

**Comments to the Author**

1. If the authors have adequately addressed your comments raised in a previous round of review and you feel that this manuscript is now acceptable for publication, you may indicate that here to bypass the “Comments to the Author” section, enter your conflict of interest statement in the “Confidential to Editor” section, and submit your "Accept" recommendation.

Reviewer #2: (No Response)

2. Is the manuscript technically sound, and do the data support the conclusions?

Reviewer #2: Yes

3. Has the statistical analysis been performed appropriately and rigorously? 

Reviewer #2: Yes

4. Have the authors made all data underlying the findings in their manuscript fully available?

Reviewer #2: No

5. Is the manuscript presented in an intelligible fashion and written in standard English?

Reviewer #2: Yes

6. Review Comments to the Author

Reviewer #2: I appreciate the authors’ revisions, especially the inclusion of the models analyzing the graduate students and postdocs/assistant professors separately. I think those analyses enhance the insights from the study considerably. I have just a few minor additional comments and suggestions.

Line 47: Please specify two or three of the “several” outcomes that were more negative for women, primary caregivers, underrepresented racial minorities, sexual minorities, and first-generation scholars. Otherwise this sentence is unnecessarily vague for an abstract

Line 121: While I appreciate the authors’ argument for the continued use of the word “prestige” in their response to reviews, it remains an inaccurate descriptor of these different institutions and unnecessarily implies author bias in importance and value of the different institutions surveyed in this study. I strongly recommend using the phrase “2011 National Research Council S-rankings”, since that is what the authors did, and is more transparent. “Prestige” is not a neutral term, and implies author bias to the reader that I think is unintended by the authors.

Table 2: I am not sure “life problems” is the best term to use here. Perhaps life disruptions, or life challenges, the term the authors use in the discussion? Or something that doesn’t qualify physical health, mental health, or caretaking as “problems”?

Line 375: replace “the results above” with “Our results”. All of the study results indicate a wide array of consequences of the COVID-19 pandemic, not just the final paragraphs and tables of the results section.

Line 415-417: It might be worth mentioning the anti-Asian sentiment in the US during 2021, as well as historic anti-Asian sentiment in the US (such as the Chinese Exclusion Act), when discussing the complex outcomes for Asian identifying scholars.

For the figures: Since the columns for the three model sets (full, graduate student, and postdoc/assistant professor) are labeled, it is unnecessary and distracting to change the pattern and color. I suggest the authors keep the colors/pattern the same

7. PLOS authors have the option to publish the peer review history of their article (what does this mean?). If published, this will include your full peer review and any attached files.

Reviewer #2: No

---

## [Author Response · Author response to Decision Letter 1]

22 Aug 2022

Response to Reviewers

Reviewer #2: I appreciate the authors’ revisions, especially the inclusion of the models analyzing the graduate students and postdocs/assistant professors separately. I think those analyses enhance the insights from the study considerably. I have just a few minor additional comments and suggestions.

Thank you for your thoughtful review and suggestions on this paper.

Line 47: Please specify two or three of the “several” outcomes that were more negative for women, primary caregivers, underrepresented racial minorities, sexual minorities, and first-generation scholars. Otherwise this sentence is unnecessarily vague for an abstract.

We included physical health symptoms and caretaking as examples in that section of the abstract.

Line 121: While I appreciate the authors’ argument for the continued use of the word “prestige” in their response to reviews, it remains an inaccurate descriptor of these different institutions and unnecessarily implies author bias in importance and value of the different institutions surveyed in this study. I strongly recommend using the phrase “2011 National Research Council S-rankings”, since that is what the authors did, and is more transparent. “Prestige” is not a neutral term, and implies author bias to the reader that I think is unintended by the authors.

We have changed the language on lines 121 and 160.

Table 2: I am not sure “life problems” is the best term to use here. Perhaps life disruptions, or life challenges, the term the authors use in the discussion? Or something that doesn’t qualify physical health, mental health, or caretaking as “problems”?

Thank you for this suggestion, we changed the title to “life challenges”

Line 375: replace “the results above” with “Our results”. All of the study results indicate a wide array of consequences of the COVID-19 pandemic, not just the final paragraphs and tables of the results section.

Thank you, we changed the language there.

Line 415-417: It might be worth mentioning the anti-Asian sentiment in the US during 2021, as well as historic anti-Asian sentiment in the US (such as the Chinese Exclusion Act), when discussing the complex outcomes for Asian identifying scholars.

Thank you for this suggestion. We’ve added that additional context to the discussion.

For the figures: Since the columns for the three model sets (full, graduate student, and postdoc/assistant professor) are labeled, it is unnecessary and distracting to change the pattern and color. I suggest the authors keep the colors/pattern the same

We opted to retain the pattern in the figures as they connect to the pattern in Tables 1 and 2. However, if the editor feels the pattern should be removed, we are happy to do so.

---

## [Editor Report · Decision Letter 2]

25 Aug 2022

Disproportionate impacts of COVID-19 on marginalized and minoritized early-career academic scientists

PONE-D-22-07647R2

Dear Dr. Montgomery,

We’re pleased to inform you that your manuscript has been judged scientifically suitable for publication and will be formally accepted for publication once it meets all outstanding technical requirements.

Kind regards,

Quinn Grundy, PhD, RN

Academic Editor

PLOS ONE
---

## [Editor Report · Acceptance letter]

1 Sep 2022

PONE-D-22-07647R2 

Disproportionate impacts of COVID-19 on marginalized and minoritized early-career academic scientists 

Dear Dr. Montgomery:

I'm pleased to inform you that your manuscript has been deemed suitable for publication in PLOS ONE. Congratulations! Your manuscript is now with our production department. 

Kind regards, 

on behalf of

Dr. Quinn Grundy 

Academic Editor

PLOS ONE